# Investigations on Ozone-Based and UV/US-Assisted Synergistic Digestion Methods for the Determination of Total Dissolved Nitrogen in Waters

**Xiaofang Sun, Huixuan Chen, Zhengyu Liu, Mengfei Zhou** 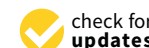**, Yijun Cai, Haitian Pan and Luyue Xia ***

College of Chemical Engineering, Zhejiang University of Technology, Hangzhou 310014, China; zgdsxf@zjut.edu.cn (X.S.); ZjutChx@126.com (H.C.); zjutlzy0811@126.com (Z.L.); mfzhou@zjut.edu.cn (M.Z.); hgybcyj@zjut.edu.cn (Y.C.); htpan@zjut.edu.cn (H.P.)

* Correspondence: lyxia@zjut.edu.cn; Tel.: +86-0571-88320329

**Abstract:** Over the past two decades, the alkaline persulfate oxidation (PO) with thermal and/or ultraviolet (UV) assisted digestion method has been widely used for digestion of nitrogen containing compounds (N-compounds) in water quality routine analysis in laboratory or on-line analysis, due to its simple principle, high conversion rate, high percent recovery, low-cost. However, this digestion method still has some inevitable problems such as complex operations, high contamination potential, batch N blanks, higher reaction temperature (120–124 °C) and time-consuming (30–60 min). In this study, ozone ($O_3$) was selected as the oxidant for digestion of N-compounds through analysis and comparison firstly. Secondly, we proposed and compared the UV and/or ultrasound (US) combined with ozone ($UV/O_3$, $US/O_3$ and $UV/US/O_3$) synergistic digestion methods based on $O_3$ with sole $O_3$ oxidation method on digestion efficiency (digestion time and conversion rate) of standard N-compounds. Simultaneously, the influence of reaction temperature, pH of water sample, concentration of $O_3$ and mass flow rate, UV intensity, US frequency and power on digestion efficiency were investigated, and then the optimum parameters for digestion system were obtained. Experimental results indicated that UV radiation can effectively induce and promote the decomposition and photolysis of $O_3$ in water to generate hydroxyl radicals (•OH), while US can promote the diffusion and dissolution of $O_3$ in water and intensify the gas-liquid mass transfer process for the reaction system. Meanwhile, results showed that the $UV/US/O_3$ synergistic digestion method had the best digestion efficiency under the optimum conditions: water sample volume, 10 mL; pH of water sample, 11; $O_3$ mass flow rate, 3200 mg/h; reaction temperature, 30 °C; digestion time, 25 min; UV lamp power, 18 W; distance between UV lamp and reactor, 2 cm; US frequency, 20 kHz; US power, 75 W. The conversion rate (CR) of synthetic wastewater samples varied from 99.6% to 101.4% for total dissolved nitrogen (TDN) in the range of 1.0~4.0 mg/L. The $UV/US/O_3$ synergistic digestion method would be an effective and potential alternative for digestion of N-compounds in water quality routine analysis in laboratory or on-line analysis.

**Keywords:** total dissolved nitrogen; digestion method; digestion efficiency; intensification; ozone

## 1. Introduction

Total nitrogen (TN) is an important indicator for water quality determination and on-line monitoring [1]. The excess of nitrogen in natural waters may cause eutrophication, which is one of the most serious threats to the aquatic ecosystems, and even do harm to human healthy [2].

Total nitrogen refers to the sum of total dissolved nitrogen (TDN) and suspended sediment nitrogen (SSN) in water sample [3]. In which, the TDN defined as the dissolved species that can passes

through a 0.2 or 0.45 μm membrane filter, and the SSN defined as the particulate fraction is retained by the filter [4]. However, determination of TDN has always been a challenging task and contemporary strategies still rely on methods which have some limitations.

Generally, TDN consists of two fractions: an inorganic fraction (i.e., dissolved inorganic nitrogen-DIN), composed of dissolved free ammonia ($NH_3$), ammonium ($NH_4^+$), nitrite ($NO_2^-$), nitrate ($NO_3^-$); an organic fraction (i.e., dissolved organic nitrogen-DON), the composition of which is unknown, but which usually include urea, proteins, amino acid, acylamide, fulvic acid and humic acid, etc. In order to determine and quantify TDN in waters, all N-compounds should be converted to a single species, such as ammonia, ammonium, nitrite and nitrate. In general, ammonia and ammonium can be distilled off in the form of ammonia and determined by titration with standardized mineral acid, nitrate and nitrite (need reduction reaction, diazo-coupling reaction and Griess reaction) can be measured by various techniques [5], but the most widely used and convenient methods are spectroscopic methods [6], such as the ultraviolet (UV)/Vis spectrophotometric method. Therefore, the first step for the determination of TDN is the digestion of the water sample in order to convert DON into an inorganic form of nitrogen, it is the most tedious and time-consuming step and the greatest source of errors in the analytical determination of TDN.

Acid Kjeldahl digestion is a traditional digestion method, which simply involves heating and boiling the sample in the presence of concentrated sulfuric acid and a mercury or copper catalyst at high temperature for several hours causing the mineralization of organic nitrogenous species to ammonium [7]. However, this digestion method involves issues of use strong acid and toxic metal catalysts, time-consuming, expensive, and only measures organic nitrogen, free ammonia and ammonium, therefore the separate determination of nitrite and nitrate by another method was required in order to determine TDN. UV photo-oxidation, first reported in 1966 by Armstrong et al. [8], which utilizes UV radiation as a source to generate strongly oxidizing hydroxyl radicals with 30% hydrogen peroxide ($H_2O_2$) as the oxidant. The main limitations of this method are that it is highly variable, has low oxidation efficiency and poor percent recovery. A wet digestion method was introduced by Koroleff who converted N-compounds to nitrate in aqueous alkaline persulfate (PS, $S_2O_8^{2-}$) [9]. Persulfate has been used and favored in flow analysis applications for it produces fewer bubbles upon decomposition compared with hydrogen peroxide. If the digestion is performed under alkaline conditions, then the sole mineral species generated is nitrate. The United States Geological Survey (USGS) states that alkaline persulfate digestion is less toxic, more accurate and sensitive to Kjeldahl digestion for nitrogen determination in environmental samples [10]. Another digestion method for TDN was later introduced with high-temperature oxidation (HTO) or named high temperature combustion (HTC) [11–13]. In this method, N-compounds were converted into •NO under high temperature more than 950 °C, and then reacted with $O_3$ to form $NO_2^*$ and emitted the photon which could be detected with a chemiluminescent detector, which needs a small amount of sample, but requires expensive equipment and is not suitable for field analysis and on-line monitoring.

Within these four methodologies, alkaline persulfate digestion remains the most mature and popular method for determination of TDN in natural waters [14–17]. This approach does not need expensive instruments, is easy to implement, and can be employed for simultaneous determination of total nitrogen and total phosphorus [18–20]. During the last 25 years, some published procedures using alkaline persulfate include batch mode [21] and flow analysis methods [22–25] with both thermal and UV radiation assisted digestion [26,27]. Flow analysis methods with UV mineralization allow high sample throughput, but the limited digestion time may result in partial conversion with interference of unreacted persulfate on the Cd/Cu column which is commonly employed in nitrate analyzers. Nevertheless, thermal digestion in batch mode is more reliable and easier controlled. In which, reaction is performed at 121 °C for 30–60 min in $H_3BO_3$/NaOH buffer at pH of 9.7 [28]. In the past two decades, the alkaline persulfate oxidation (PO) with thermal or UV (UV/PO) has been widely used in total nitrogen routine analysis in laboratory or on-line analysis due to its simple principle, high conversion rate, high percent recovery, and low-cost. Nowadays, total nitrogen (TN) on-line monitors are widely

sold in the international market and used in environmental monitoring, such as famous NPW-160H (HACH), TNP-4200 (SHIMADZU) and WPA-58 (KEM), which are basically used for the alkaline persulfate oxidation with thermal or UV assisted digestion method.

However, this method still has some inevitable problems such as complex operations, high contamination potential, batch N blanks [29], higher reaction temperature (120–124 °C) and time-consuming (30–60 min). As we know, UV/PO and UV/$H_2O_2$ are the advanced oxidation processes (AOPs) [30,31] based on digestion methods, in which the oxidant is the key of digestion, under UV radiation, PS can generate strongly reactive yet selective sulfate radicals ($\bullet SO_4^-$), and $H_2O_2$ can generate hydroxyl radicals ($\bullet OH$) with super oxidation capacity, which can react with the majority of organic compounds and degrade the contaminants in waters non-selectively [32]. According to the commonly used oxidants in water treatment [33], besides the PS and $H_2O_2$, ozone ($O_3$) may be a compromised alternative for N-compounds digestion, and which can generate $\bullet OH$ by self-decomposition in basic medium or under certain conditions such as UV, ultrasound (US) and microwave (MW) [34,35]. Although ozone has been widely used in water treatment as disinfectant and oxidant, in which transformation of organic micropollutants occurs via direct reaction with $O_3$ or indirect reaction with hydroxyl radicals [36–40], fewer studies have reported its use for digestion and determination of TDN in waters. Considering the oxidation capacity, safety, self-degradability, removal ability and convenience of oxidant, $O_3$ was selected as the oxidant for digestion of N-compounds, and then will use UV and/or US to intensify its digestion ability in this study.

With this background, the main objectives of this study were: (a) to compare the UV/$O_3$, US/$O_3$ and UV/US/$O_3$ synergistic digestion methods based on $O_3$ with sole $O_3$ oxidation method on digestion efficiency (digestion time and conversion rate) of standard N-compounds; (b) to investigate the influence of reaction temperature, pH of water sample, concentration of $O_3$ and mass flow rate, UV intensity, US frequency and power on digestion efficiency, and then to obtain the optimum parameters for digestion system; (c) to seek and develop a novel and reliable digestion method for safe, environmentally-friendly, convenient and efficient digestion of nitrogen containing organic and inorganic compounds for improving the precision and stability of determination of total dissolved nitrogen in natural waters and waste waters. Section 2 presented the novel methodology for determination of TDN, including the reagents, solutions, apparatus, experimental devices, and detailed procedure. Section 3 consisted of experimental results and discussion. Section 4 addressed the conclusions of this research and directions for future work.

## 2. Materials and Methods

### 2.1. Reagents and Solutions

All chemicals used in experiments were of analytical grade and superior grade, all solutions were prepared with nitrogen-free water. The N-compounds used in this study included representative dissolved inorganic nitrogen (DIN) and dissolved organic nitrogen (DON), such as potassium nitrate, sodium nitrate, ammonia chloride and carbamide, which are frequently occurring components in natural waters or waste waters from factories. All stock solutions of them were prepared with nitrogen concentration of 100 mg/L, then all working standard solutions were prepared by diluting from the stock solutions, such as 10 mg/L, 4 mg/L and 1 mg/L. In addition, the concentration of sodium hydroxide and hydrochloric acid solutions used to regulate the pH of water sample were 0.01 mol/L.

### 2.2. Apparatus and Devices

In order to establish a suitable digestion method for N-compounds in waters, different process intensification methods (UV, US, and combination of UV & US) were proposed and investigated in this study.

In the first process intensification scheme, two UV lamps were used to assist $O_3$ for oxidation of N-compounds, and the schematic diagram of UV/$O_3$ synergistic digestion experimental device is presented in Figure 1.

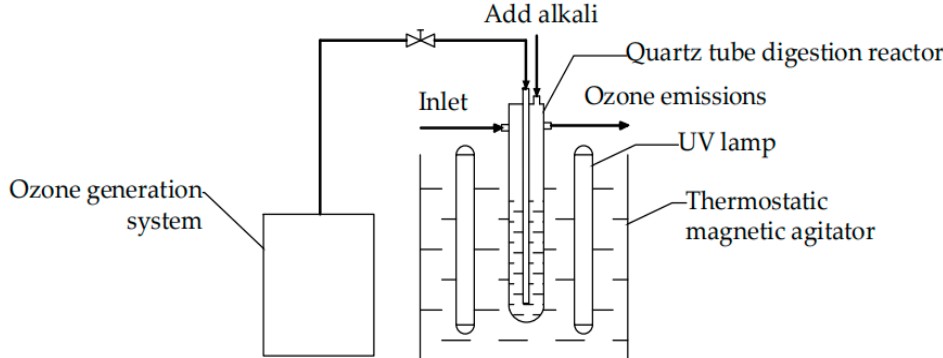

**Figure 1.** Schematic diagram of UV/$O_3$ synergistic digestion experimental device.

As shown in Figure 1, two UV lamps hanged around the quartz tube digestion reactor are 18 W low-press mercury lamps with maximum emission at 254 nm (Cnlight Co., Guangdong, China), and the jacket of the lamp was made up of quartz for strong and sharp emission. In this device, the quartz tube digestion reactor is a straight and cylindrical reactor having 12 mm inner diameter (i.d.) and 180 mm height. A 1–10 mL adjustable micropipette (Lichen Instruments Co., Shanghai, China) was used to transfer the working standard solutions and reagents. The self-made electrolytic ozone generation system was consisted of four DJ800 polymer electrolysis membrane (PEM) electrode modules, two 3–5 V/12 A (w) power modules and four water tanks, in which the mixed gas of ozone/oxygen with a mass concentration approximately of 18–20 wt/wt% (250–280 mg/L, 3 L/h) were prepared and gathered in two anode water tanks by electrolyzing deionized water, two cathode water tanks gathered the by-products $H_2$. In this system, two PEMs, two water tanks and one power module form one ozone generator, and two generators run in parallel. In addition, in order to keep reaction temperature in constant, the DF-101S thermostatic magnetic agitator (Lichen Instruments Co., Shanghai, China) was used to control the digestion reaction temperature (±0.1 °C).

The second scheme used US to assist $O_3$ for oxidation of N-compounds, and the schematic diagram of US/$O_3$ synergistic digestion experimental device is presented in Figure 2.

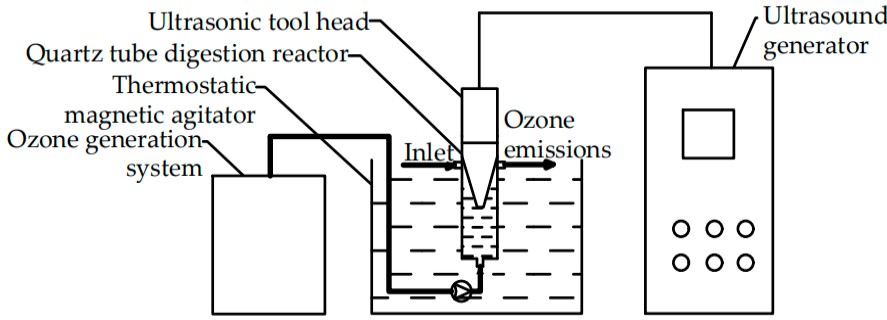

**Figure 2.** Schematic diagram of US/$O_3$ synergistic digestion experimental device.

As shown in Figure 2, the ultrasound generator with 10 mm ultrasonic probe was used and the optimum working power range of 50–100 W. Considering the convenience of installation for ultrasonic probe and keep it under the surface of digested solution, the quartz tube digestion reactor was designed with 20 mm inner diameter (i.d.) and 80 mm height. In addition, the ozone generation system and thermostatic magnetic agitator were used same as scheme one in this device.

In the third scheme, UV in combination with US digestion method was used to assist and intensify the digestion ability of $O_3$, in which the UV/US/$O_3$ synergistic digestion experimental device was applied.

Accordingly, an 754PC ultraviolet-visible spectrophotometer (Jinghua Instruments Co., Shanghai, China) and two quartz colorimetric cells with light path of 10 mm were used for TDN determination in detection system.

### 2.3. Procedure and Methods

The detailed procedure was as follows:

(1) Turn on the ozone generation system and close the outlet valve of $O_3$ synchronously;

(2) Turn on the thermostatic magnetic agitator and control the digestion reaction temperature at 30 °C;

(3) Transfer 9 mL of standard solution or water sample of carbamide and 1 ml of sodium hydroxide solution (0.01 mol/L) to the quartz tube digestion reactor by adjustable micropipette;

(4) Turn on the UV lamps (scheme 1 and scheme 3), US generator (scheme 2 and scheme 3) and open the outlet valve of $O_3$, high concentration of ozone gas bubbled through an ozone bubble tube into the bottom of digestion reactor for digestion of N-compounds;

(5) Several minutes later (5, 10, 15, 20 and 25 min), turn off the UV lamps, US generator and ozone generation system;

(6) Heat the digested standard solution or water sample to 50 °C and drive out the dissolved $O_3$ for 5 min;

(7) Transfer 5 mL of digested standard solution or water sample to the quartz colorimetric cell for the detection of absorbance at 220 nm and 275 nm, respectively;

(8) Calculate the concentrations of water samples using the prepared calibration curve, according to the net absorbances calculated by subtracting the corrected absorbances of digested blank solutions from the corrected absorbances of digested standard solutions.

In this procedure, the calibration curve was prepared by nitrate standard solutions and went through the whole digestion and detection procedure, in which the concentration of TDN was in the range of 0.5~4.0 mg/L and the mean N blank is 0.020 in 14 blank tests. The calibration curve of TN is shown in Figure 3.

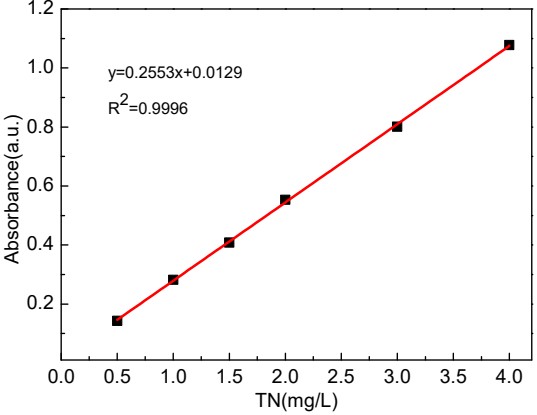

**Figure 3.** Calibration curve of TN.

Moreover, the percent conversion of N-compounds to their nitrate forms were calculated with respect to the signal generated using standards of potassium nitrate at an equivalent nitrogen concentration.

## 3. Results and Discussion

### 3.1. UV/O₃ Digestion

According to the basic principles of chemical reaction kinetics, the concentration of oxidant dissolved in water is a very important factor for the gas-liquid heterogeneous reaction system of $O_3$ with N-compounds. By Henry's law, under certain temperature conditions, the higher the concentration of $O_3$ in gas phase, the higher those in aqueous solution. Consequently, improving the concentration of $O_3$ in gas phase and the quantity of ozone gas are important to intensify the ozone mass transfer process. On the other hand, ozone dissolved in aqueous solution can self-decompose and form the intermediate products of hydrogen peroxide, which can be motivated by the energy of photon to generate ●OH under the condition of ultraviolet radiation [41]. Meanwhile, the photolysis of ozone can generate highly active oxygen radicals (●O), and then react with water to form ●OH [42]. So the digestion efficiency of UV/$O_3$ is mainly affected by UV radiation and $O_3$ concentration dissolved in water, for which the electrolytic ozone generation system with 4 PEM modules was constructed and the ozone gas with a mass concentration approximately of 18–20 wt/wt% was prepared in this study.

In order to investigate the effect of reaction temperature on digestion efficiency, the univariate experiments were conducted by increasing temperature, in which the digested nitrogen containing compound was the carbamide solution with concentration of 10 mg/L and without the addition of sodium hydroxide, and the reaction time was 10 min. Results showed that the digested solution had the highest absorbance and the best conversation rate when the reaction temperature at 30 °C. As we known, although the saturation concentration of ozone in liquid phase is higher at lower temperature, but the reaction rate constant and the rate of ozone self-decomposition are lower, so the efficiency of ozone conversion to ●OH is lower too. However, when the temperature is higher than 35 °C, although the reaction rate is faster, the low concentration of ozone in solution leads to the decline of the probability of contact between ozone and reactant, and the digestion efficiency declines spontaneously. In the actual digestion process, the reaction temperature may be higher than 30 °C due to the long time radiation of UV lamps. Therefore, it is very important to avoid the decline of digestion efficiency by dissipating the heat of digestion system.

For the UV intensity, according to the Lambert law, UV lamp power and distance between UV lamp and reactor are the main two factors affected the UV intensity and photolysis rate. For the convenience of investigation, the distance between UV lamp and reactor (2 cm) was fixed at first, and then three types of UV lamps with different power (15 W, 18 W and 23 W) were used and investigated under the conditions of carbamide solution with concentration of 1 mg/L, reaction temperature of 30 °C, without the addition of sodium hydroxide. Results are shown in Figure 4.

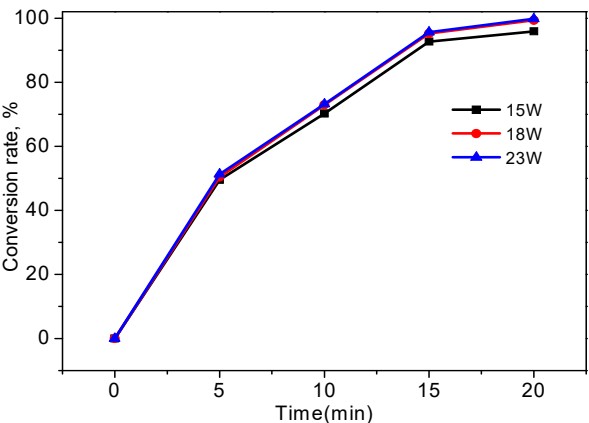

**Figure 4.** Effect of ultraviolet (UV) intensity on digestion efficiency.

As shown in Figure 4, the digestion efficiency was found to be increasing with the increased power, and no significant changes were observed at power higher than 18 W. Mainly due to the facy

that the UV lamp of 18 W has enough energy and photons to induce and motivate $O_3$ to photolyze and generate •OH, while the 15 W UV lamp is relatively weaker and lower. By using 18 W or 23 W UV lamp, the carbamide solution with concentration of 1 mg/L could be completely digested in 20 min. Consequently, the 18 W UV lamp was selected to be used in the subsequent experiments.

For the $UV/O_3$ synergistic digestion scheme, pH has a great influence on the conversion of •OH in the reaction system. Under acidic conditions, only about 20% of the ozone in aqueous solution decomposes directly to •OH, whereas under alkaline conditions, the photolysis of ozone and hydroxyl ions ($OH^-$) can motivate the formation of •OH, and the conversion rate of •OH is much higher [43,44]. Experimental results are shown in Figure 5.

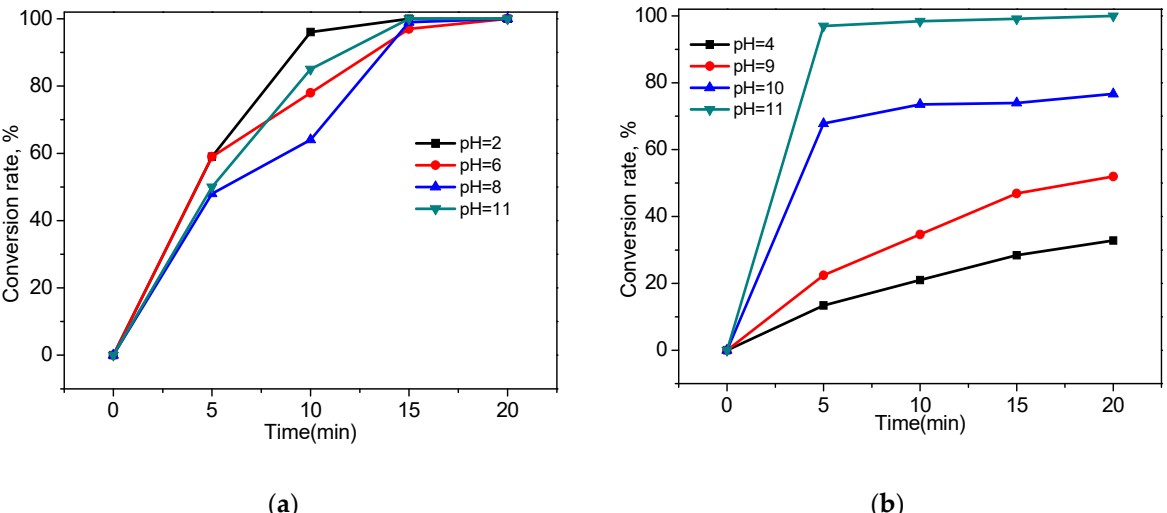

(**a**)                                             (**b**)

**Figure 5.** Effect of pH on digestion efficiency. (**a**) Digestion of carbamide solution with concentration of 1 mg/L; (**b**) Digestion of ammonia chloride solution with concentration of 1 mg/L.

It can be seen from Figure 5a that the carbamide can be quickly digested under both alkaline and acidic conditions, and the carbamide has the highest digestion efficiency with the pH at 2, which mainly due to the higher oxidation-reduction potential of ozone under acidic conditions ($E^0$ = 2.07 V) than that under alkaline conditions ($E^0$ = 1.24 V). Whereas, $OH^-$ can promote the generation of •OH under alkaline conditions, so that the synergistic effect of the reaction process is enhanced [45] when pH at 11. When the pH is at 6, reactions remained the same conversion rate as that of pH at 2 in the first 5 min, but it declined quicker in the rest of reaction time. It may be explained that: under acidic condition and UV radiation, $O_3$ dissolved in aqueous solution can form the intermediate products of $H_2O_2$, which can be motivated by UV to generate •OH, while the ionization of $H_2O_2$ was greatly influenced by pH of solution, the higher pH of solution is, the more $H_2O_2$ will be ionized, and the less •OH will be generated by photolysis of $H_2O_2$.

As shown in Figure 5b, the digestion process of ammonia chloride is greatly influenced by pH of solution. When the pH at 11, the reaction rate constant of •OH with $NH_3$ is about $1.7–8.7 \times 10^7$ $M^{-1} \cdot S^{-1}$, so that the digestion efficiency is very high and the digestion time need only 5 min. However, when the pH is at 10, the reaction rate in first 5 min is high, and after that it declines rapidly and remains basically unchanged, which is mainly due to the consumption of $OH^-$ and its rapid decline of concentration. As the ozone and •OH are difficult to react with $NH_4^+$ [46], the weak conversion rate of ammonia chloride was obtained with the pH at 4. Considering the ammonia nitrogen and carbamide nitrogen coexist in the natural waters or waste waters, high pH value (pH = 11) was selected to be used in this study.

To further verify the effectiveness of $UV/O_3$ digestion method under alkaline conditions, comparative experiments were carried out based on the above results obtained, and the results are shown in Figure 6.

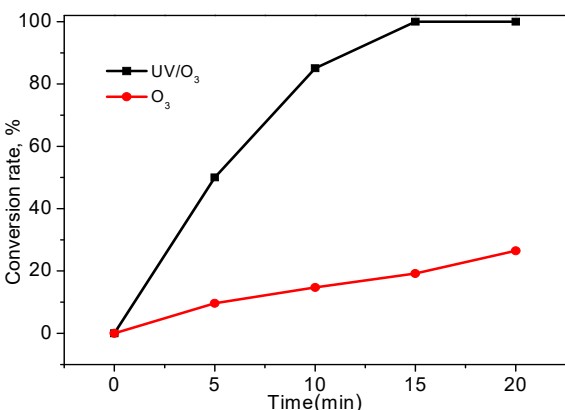

**Figure 6.** Comparison of digestion efficiency between UV/$O_3$ and $O_3$.

Compared with UV/$O_3$ digestion method, weak percent conversion was obtained by sole $O_3$ digestion method for carbamide digestion with concentration of 1 mg/L and the UV radiation had excellent intensification effect on digestion process of $O_3$, which can effectively induce and promote the radical reaction of $O_3$ in aqueous solution to generate •OH with super oxidation capacity.

### 3.2. US/$O_3$ Digestion

In the US/$O_3$ synergistic digestion scheme, the intensification mechanisms of US are mainly reflected in two aspects, one is to intensify the mass transfer of ozone, the other is to promote the generation of •OH. Under certain frequency and sound intensity, US can crush the lager ozone bubbles into microbubbles with a diameter of about 0.2–0.3 μm, and the total contact area of gas-liquid will be greatly improved. Moreover, the gas-liquid mass transfer coefficient will be greatly enhanced by strong mechanical stirring effect of US. On the other hand, the local high temperature and high pressure environment caused by ultrasound cavitation effect will also promote the generation of •OH [47–49].

In this study, the ultrasound frequency of 20 kHz and power of 75 W were chosen as optimizing parameters for digestion of TDN as suggested by the previous literature [50]. In order to verify the effectiveness of US/$O_3$ digestion method, comparative experiments were conducted based on the reaction conditions obtained from above results, and carbamide solution with concentration of 100 mg/L. Results are shown in Figure 7.

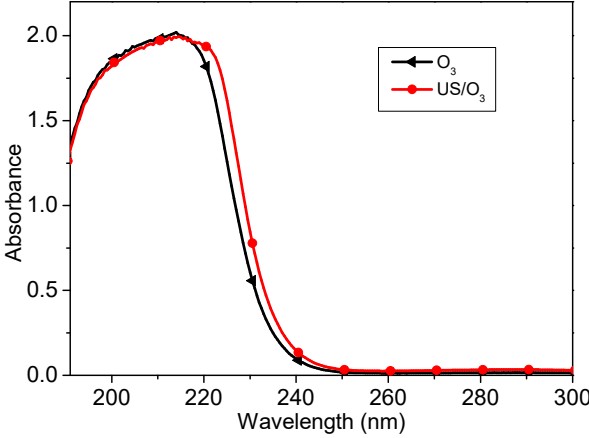

**Figure 7.** Comparison of digestion efficiency between US/$O_3$ and $O_3$.

Compared with sole $O_3$ digestion method, the intensification effect of US on digestion process of $O_3$ is not obvious, which is mainly due to the low conversion rate of •OH, meanwhile, the •O generated by US could be consumed by $O_3$. On the other hand, although the US can promote the

diffusion and dissolution of $O_3$ in water, the lower level (32 mm) of digested solution in the reactor of $US/O_3$ synergistic digestion experimental device led to the shorter residence time of ozone in aqueous solution, and the reaction time with N-compounds was shortened too.

### 3.3. UV/US/O$_3$ Digestion

Combining the advantages of UV, US and $O_3$, comparative experiments of $UV/US/O_3$ and $UV/O_3$ digestion methods were conducted based on the optimal conditions, in which the concentration of carbamide solution is 4 mg/L. Results are displayed in Figure 7.

As shown in Figure 8, the $UV/US/O_3$ synergistic digestion method had an excellent digestion efficiency, and the carbamide solution with concentration of 4 mg/L could be completely digested in 25 min. Meanwhile, it can be seen that ultrasound can effectively promote the diffusion and dissolution of $O_3$ in aqueous solution and intensify the gas-liquid mass transfer for reaction system.

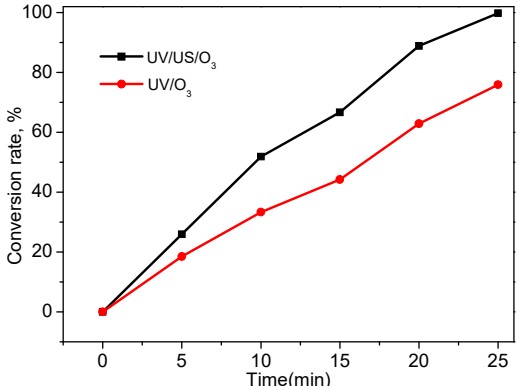

**Figure 8.** Comparison of digestion efficiency between $UV/US/O_3$ and $UV/O_3$.

### 3.4. Optimization of UV/US/O$_3$ Process

The optimization of the $UV/US/O_3$ digestion process was conducted in order to obtain feasible, economical and maximum parameters for percent conversion of TDN. In our work, the orthogonal experimental design was applied, in which a six-factor (four real factors and two dummy factors) and five-level $L_{25}$ ($5^6$) orthogonal table was employed to investigate the effects of four independent factors (distance between UV lamp and reactor (A), US power (B), water sample volume (C), digestion time (D)) on conversion rate (CR) based on the preliminary experimental results. The orthogonal experiment design and the results of orthogonal experiment are shown in Tables 1 and 2, respectively.

**Table 1.** Orthogonal experimental design.

| Factor | A (cm) | B (W) | C (mL) | D (min) |
|---|---|---|---|---|
| Level 1 | 1 | 50 | 10 | 10 |
| Level 2 | 2 | 62.5 | 15 | 15 |
| Level 3 | 3 | 75 | 20 | 20 |
| Level 4 | 4 | 87.5 | 25 | 25 |
| Level 5 | 5 | 100 | 30 | 30 |

As shown in Table 2, twenty five experiments were run under the conditions of the carbamide solution with concentration of 4 mg/L, $O_3$ mass flow rate of 3200 mg/h, reaction temperature of 30 °C, pH at 11, UV lamp power of 18 W and US frequency of 20 kHz, which was fixed from the previous study.

Where Ki refers to the average value of five conversion rates at the i level of each factor, which reflects the influence of each level of each factor on the conversion rate, the larger Ki value, the higher CR at the i level. While Range is the difference between the maximum and minimum Ki values, which reflects the degree of influence of various factors on CR, the larger R value, the more important this factor.

Therefore, the main influencing factors of the UV/US/O$_3$ digestion process were ranked D > B > C > A, and the recommended optimal combination was A$_2$B$_3$C$_1$D$_4$. Consequently, the optimized parameters were obtained by range analysis, viz.: distance between UV lamp and reactor, 2 cm; US power, 75 W; water sample volume, 10 mL; digestion time, 25 min, according to the actual values of different factor corresponding to each level displayed in Table 1.

**Table 2.** Results of orthogonal experiment.

| Experiment No. | A | B | C | D | Dummy | Dummy | CR (%) |
|---|---|---|---|---|---|---|---|
| 1 | 1 | 1 | 1 | 1 | 1 | 1 | 26.5 |
| 2 | 1 | 2 | 2 | 2 | 2 | 2 | 40.0 |
| 3 | 1 | 3 | 3 | 3 | 3 | 3 | 64.2 |
| 4 | 1 | 4 | 4 | 4 | 4 | 4 | 80.3 |
| 5 | 1 | 5 | 5 | 5 | 5 | 5 | 75.0 |
| 6 | 2 | 1 | 2 | 3 | 4 | 5 | 57.2 |
| 7 | 2 | 2 | 3 | 4 | 5 | 1 | 71.9 |
| 8 | 2 | 3 | 4 | 5 | 1 | 2 | 93.7 |
| 9 | 2 | 4 | 5 | 1 | 2 | 3 | 12.4 |
| 10 | 2 | 5 | 1 | 2 | 3 | 4 | 50.9 |
| 11 | 3 | 1 | 3 | 5 | 2 | 4 | 69.6 |
| 12 | 3 | 2 | 4 | 1 | 3 | 5 | 21.4 |
| 13 | 3 | 3 | 5 | 2 | 4 | 1 | 26.1 |
| 14 | 3 | 4 | 1 | 3 | 5 | 2 | 77.0 |
| 15 | 3 | 5 | 2 | 4 | 1 | 3 | 77.4 |
| 16 | 4 | 1 | 4 | 2 | 5 | 3 | 37.8 |
| 17 | 4 | 2 | 5 | 3 | 1 | 4 | 40.4 |
| 18 | 4 | 3 | 1 | 4 | 2 | 5 | 97.4 |
| 19 | 4 | 4 | 2 | 5 | 3 | 1 | 74.8 |
| 20 | 4 | 5 | 3 | 1 | 4 | 2 | 18.9 |
| 21 | 5 | 1 | 5 | 4 | 3 | 2 | 89.5 |
| 22 | 5 | 2 | 1 | 5 | 4 | 3 | 52.6 |
| 23 | 5 | 3 | 2 | 1 | 5 | 4 | 18.5 |
| 24 | 5 | 4 | 3 | 2 | 1 | 5 | 35.8 |
| 25 | 5 | 5 | 4 | 3 | 2 | 1 | 46.6 |
| K$_1$ | 57.20 | 56.12 | 60.88 | 19.54 | | | |
| K$_2$ | 57.22 | 45.26 | 53.58 | 38.12 | | | |
| K$_3$ | 54.30 | 59.98 | 52.08 | 57.08 | | | |
| K$_4$ | 53.86 | 56.06 | 55.92 | 83.30 | | | |
| K$_5$ | 48.60 | 53.76 | 48.68 | 73.14 | | | |
| Range | 8.62 | 14.72 | 12.20 | 63.76 | | | |
| Rank | | D > B > C > A | | | | | |
| Optimal combination | | A$_2$B$_3$C$_1$D$_4$ | | | | | |

*3.5. Validation of Digestion Efficiency*

In order to further evaluate the digestion efficiency of the proposed method under the optimum conditions obtained from orthogonal experiment. The carbamide solution with concentration ranging from 1.0 to 4.0 mg/L were digested and analyzed, each group was run in triplicate, results are displayed in Table 3. As shown in Table 3, where A$_s$ represents the final absorbance of standard solution of sodium nitrate digested under optimum conditions, A$_1$, A$_2$ and A$_3$ represent the final absorbance of carbamide solution digested under the same conditions in triplicate, respectively, and A$_m$ represents the mean absorbance.

**Table 3.** Validation of digestion efficiency.

| TDN (mg/L) | $A_s$ | $A_1$ | $A_2$ | $A_3$ | $A_m$ | CR (%) |
|---|---|---|---|---|---|---|
| 1.0 | 0.282 | 0.290 | 0.287 | 0.291 | 0.289 | 102.48 |
| 2.0 | 0.543 | 0.540 | 0.536 | 0.537 | 0.538 | 99.08 |
| 3.0 | 0.806 | 0.799 | 0.796 | 0.803 | 0.799 | 99.13 |
| 4.0 | 1.073 | 1.091 | 1.052 | 1.037 | 1.060 | 98.79 |

It can be seen in Table 3 that the percent conversion of TDN ranged from 102.48% to 98.79% with an increase of TDN from 1.0 mg/L to 4.0 mg/L, and the proposed digestion method had a good digestion efficiency, by which the carbamide solution with concentration of 4.0 mg/L could be completely digested in 25 min under the optimum conditions obtained in this study.

However, some interference factors during the experiments caused a certain degree of deviations in experimental results. First of all, the N blank value was the mean value of 14 blank tests in making the calibration curve of TN, while the actual N blank values deviated from the mean value, which may be positive or negative. Secondly, after the reaction, the pH of the digested solution was weakly alkaline, and the $OH^-$ in the solution had an effect on the absorbance at 220 nm. Moreover, the residual ozone dissolved in water could not be completely removed by heating the solution, which also had an effect on the absorbance at 220 nm. Finally, in the detection process, the noise of the instrument also affected the experimental results. Therefore, with the same interference of absorbance, the lower concentration of TDN in water, the higher deviation of the conversion rate.

### 3.6. Application of Digestion Method

The proposed UV/US/$O_3$ synergistic digestion method was applied to determine the TDN in a synthetic wastewater samples, which refer to the wastewater produced from urea production process [51]. Analysis results of the actual urea wastewater are shown in Table 4.

**Table 4.** Analysis results of the actual urea wastewater.

| Analysis Contents | Analysis Data |
|---|---|
| pH | 9 |
| alkalinity | 19.6 mmol/L |
| total iron | 0.49 mg/L |
| chloridion | 16 mg/L |
| ammonia chloride | 0.034 wt% |
| urea | 0.57 wt% |
| formic acid | a little |

According to the contents of actual wastewater, synthetic wastewater samples were prepared by the ratio 19:1 of urea to ammonia chloride, in which the concentration of TDN was in the range of 1.0~4.0 mg/L, other contents and its concentrations remained unchanged, and then each group was run in 6 times, the concentration of TDN was determined by UV spectrophotometric method. Determination results are displayed in Table 5.

**Table 5.** Determination results of the synthetic wastewater samples.

| TDN (mg/L) | Test 1 (mg/L) | Test 2 (mg/L) | Test 3 (mg/L) | Test 4 (mg/L) | Test 5 (mg/L) | Test 6 (mg/L) | Mean (mg/L) | RSD (%) | CR (%) |
|---|---|---|---|---|---|---|---|---|---|
| 1.0 | 1.03 | 1.01 | 1.04 | 0.98 | 1.01 | 0.99 | 1.014 | 2.09 | 101.4 |
| 2.0 | 2.04 | 1.97 | 2.02 | 2.03 | 1.99 | 1.96 | 2.010 | 1.50 | 100.5 |
| 3.0 | 2.96 | 3.06 | 2.96 | 3.01 | 2.95 | 3.02 | 2.988 | 1.33 | 99.6 |
| 4.0 | 4.03 | 4.05 | 3.97 | 4.03 | 4.09 | 3.97 | 4.034 | 1.08 | 100.9 |

In this case, it can be seen from Table 5 that the proposed UV/US/$O_3$ synergistic digestion method combined with common UV spectrophotometric detection method had good performance, relative standard deviations (RSD) and the conversion rate (CR) were in the range 1.08~2.09% and 99.6~101.4%, respectively.

## 4. Conclusions

This study is the first time that the combination of both UV radiation and US simultaneously with $O_3$ oxidation was used to digest the N-compounds for determination of total dissolved nitrogen in waters. In this work, the advantage of UV/US/$O_3$ synergistic digestion method combined with common UV spectrophotometric detection method has been clearly shown by the digestion efficient and fast determination of total dissolved nitrogen. Results of experimental investigation showed that UV can effectively induce and promote the decomposition and photolysis of $O_3$ in aqueous solution to generate ●OH, while US can intensify the gas-liquid mass transfer for reaction system, by which the carbamide solution with concentration of 4.0 mg/L could be completely digested under the optimum conditions: water sample volume, 10 mL; pH of water sample, 11; $O_3$ mass flow rate, 3200 mg/h; reaction temperature, 30 °C; digestion time, 25 min; UV lamp power, 18 W; distance between UV lamp and reactor, 2 cm; US frequency, 20 kHz; US power, 75 W. The conversion rate (CR) of synthetic wastewater samples varied from 99.6% to 101.4% for TDN in the range of 1.0~4.0 mg/L.

However, only three species of standard N-compounds (i.e., nitrate, ammonia chloride and carbamide) were investigated in our work. Natural waters may contain more species of TDN, especially for DON, the composition of which is unknown, but which usually include proteins, amino acid, acylamide, fulvic acid and humic acid, etc. In this case, more species of DON would need to be investigated. Meanwhile, in order to simulate and optimize the digestion progresses of N-compounds by UV/US/$O_3$ method, the degradation mechanism and reaction kinetics of N-compounds need for further study. Furthermore, due to the limitation of pressure and quantity of ozone generated by self-made ozone generation system, the ozone mass flow rate could not be adjusted conveniently, a novel ozone generation system based on pure oxygen would be established. Finally, a new jacketed digestion reactor need to be designed and developed for further studies and applications. Future work will focus on these issues.

**Author Contributions:** Conceptualization, X.S., L.X. and M.Z.; Methodology, X.S. and L.X.; Software, H.C.; Validation, H.C. and Z.L.; Formal analysis, Z.L.; Investigation, H.C.; Data curation, Z.L.; Writing—original draft preparation, X.S. and H.C.; Writing—review and editing, L.X.; Supervision, M.Z.; Project administration, Y.C. and L.X.; Funding acquisition, H.P. All authors have read and agreed to the published version of the manuscript.

**Funding:** This research was supported by Application Research of Public Welfare Technology in Zhejiang Province, China, Grant No. LGF19B060003 and LGF20E090005; National Natural Science Foundation of China, Grant No. 21676251.

**Acknowledgments:** The authors would also like to acknowledge everyone who has provided helpful guidance and would also like to thank the anonymous reviewers for their useful comments.

**Conflicts of Interest:** The authors declare no conflict of interest.

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
