# Peer review of "Investigations on Ozone-Based and UV/US-Assisted Synergistic Digestion Methods for the Determination of Total Dissolved Nitrogen in Waters"

_processes, doi:10.3390/pr8040490_

Round 1
Reviewer 1 Report
The paper “Investigations on Ozone-based and UV/US-assisted Synergistic Digestion Methods for the Determination of Total Dissolved Nitrogen in Waters” deals with the digestion methods for nitrogen, considering that the ultraviolet and ultrasound combined with ozone (UV/US/O3) method had the best digestion efficiency. I think authors improved the general level of this manuscript fulfilling the previous requirement and I think it deserves publication after minor changes.
Introduction:
This reviewer think that introduction section meets the proper quality of a scientific paper. The introduction section is now better described. They also identified the gaps in research (Lines 96-112) and show their key goals, as they compare the UV/O3, US/O3 and UV/US/O3 digestion methods. Authors have also included limitations of their methodology
In my opinion, they should include a small paragraph (the last in the introduction section), showing the reader where we could find paper structure
Minor remarks: Line19: I think first time you use the acronyms UV and US, you should show what they mean (to help readers). Ultrasound is written quite far from here…
Methodology:
Lines 77-80 should be removed; this is widely known. DONE, OK
Lines 85-89 and 91-100, These paragraphs should be incorporated into the introduction section. DONE, OK
Results and discussion
Figure 4: In my opinion, this figure is not interesting for the reader. Reader may not find any significant change among the UV intensity. I would also like you to have added some extra information to this fact. To name a few:
- Analysis of the Radiation Dose in UV-Disinfection Flow Reactors. Wojciech Artichowicz * Aneta Luczkiewicz and Jerzy M. Sawicki. Water 2020, 12, 231; doi:10.3390/w12010231
- : Application of UV light emitting diodes to batch and flow-through water disinfection systems Kumiko Oguma a, ⁎, Ryo Kita a , Hiroshi Sakai a , Michio Murakami b , Satoshi Takizawa. 1016/j.desal.2013.08.014
Table 2: I am not familiar with the parameters K1-K5 showed in the table. This should be explained.
I would also like to know why the combination A2B3C1D4 is better than A4B3C1D4. This should be explained.
Minor remark: Table 2 should be in Section 3.4, not in Section 3.5
Minor remark: Table 3. Lines 334-340 should be located before the Table, this would help the reader to understand As?A1?A2?A3 and Am?, Anyway, I would like to see why it is better the TDN (mg/l)=4 than the TDN=1 mg/l test. I understand that the medium absorbance is more important than CR(%), but I would like this to be explained.
I would also like to find a future developments section in this section.
Author Response
Dear reviewer,
Thanks for your good comments and suggestions for our rewritten manuscript. We had revised the manuscript carefully according to your suggestions again, and the point-by-point response to your comments was uploaded as an attachment, please review the detailed attachment again, thanks a lot!
Thank you very much for your comments and consideration!
Best regards,
Xiaofang Sun
19 Apr 2020

Reviewer 2 Report
The authors improved a lot the paper, they adequately answered at the reviewers suggestions and I appreciate that the paper could be published in the present version.
Author Response
Dear reviewer,
Thanks for your good comments and suggestions for our rewritten manuscript, thanks a lot!
Thank you very much for your consideration again!
Best regards,
Xiaofang Sun
19 Apr 2020
This manuscript is a resubmission of an earlier submission. The following is a list of the peer review reports and author responses from that submission.
Round 1
Reviewer 1 Report
The paper “Investigations on Process Intensification Methods for Total Nitrogen Digestion in Water Quality on-line Monitoring” deals with the digestion methods for nitrogen, considering that the ultraviolet and ultrasound combined with ozone (UV/US/O3) method had the best digestion efficiency. The research seems to be interesting, but some parts of the manuscript should be better described (in my opinion). That is the reason as I choose to reject this manuscript, although I think it should have the potential to be published after a huge change in it.
Remarks: Authous should clearly identify how they make science to advance. After a quick search I have found many papers regarding this fact, to name a few:
- “The Role of Ultrasound on Advanced Oxidation Processes”, Sundaram Ganesh Babu, Muthupandian Ashokkumar, Bernaurdshaw Neppolian1, DOI 10.1007/s41061-016-0072-9
- “Evaluation of a second derivative UV/visible spectroscopy technique for nitrate and total nitrogen analysis of wastewater samples” Michelle AFerree, Robert D. Shannon https://doi.org/10.1016/S0043-1354(00)00222-0
- “High-Efficiency Microwave-Assisted Digestion Combined to in Situ Ultraviolet Radiation for the Determination of Rare Earth Elements by Ultrasonic Nebulization ICPMS in Crude Oils”, J. S. F. Pereira, R. S. Picoloto L. S. F. Pereira R. C. L. Guimarães R. A. Guarnieri, E. M. M. Flores, https://doi.org/10.1021/ac402928u
- Microwave-assisted UV-digestion procedure for the accurate determination of Pd in natural waters” A Limbeck - Analytica chimica acta, 2006 – Elsevier. https://doi.org/10.1016/j.aca.2006.05.062
…
This part is essential to carry on the publication process. Authors should identify the gaps in research and show the key goals obtained.
Introduction:
This reviewer think that introduction section does not meet the proper quality of a scientific paper. The introduction section is quite low, authors should include more references and moreover, they should not use 6 references cited simultaneously (as it is in Line 42, where you cite 6-12).
Lines 43-48. This should be included in the methods and materials section.
Lines 49-55. This paragraph should be justified much better, with references, etc..
Authors should include limitations of their methodology, their key goals and overall, why they are making science to advance.
Introduction should be organised as follows:
- Gaps of knowledge
- Objective
- paper structure
Methodology:
Lines 77-80 should be removed, this is widely known.
Lines 85-89 and 91-100, These paragraphs should be incorporated into the introduction section
Results and discussion
Lines 164-166 These lines are not results, It should be considered in methods and materials.
Lines 175-179: This should be removed or moved into an appendix
Figure 3 is widely known, remove (see “Effects of Temperature and Hydraulic Retention Time on Anaerobic Digestion of Food Waste” Jung Kon Kim, Baek Rock Oh, Young Nam Chun, and Si Wouk Kim or some others)
Figure 4: This figure is not well presented. Reader may not find any significant change among the UV intensity.
Figure 5a) Authors should explain the strange behaviour of the ph=6 graph.
Lines 251-252: Formulas should be removed
The discussion section should be included. Some values comparing your results with some others from the bibliography should be included
I would also like to find a future developments section in this section.
Conclusions:
Conclusions must be improved.
Reviewer 2 Report
The paper is treating an important problem for the environment: pollutants degradation by photochemical ways.
The introduction is too short and not comprehensive for this area. The same for Conclusions, is too poor and not relevant.
The necessary details for photochemical part are missing: incident radiation, actinometry, degradation mechanism for carbamide, the contribution of each oxygen species at the carbamide degradation, rate reactions, etc.
I consider that this paper should be re-written answering at the above mentioned questions.